cognition, evolution, behaviour

sound symbolism, language evolution, Kanzi, bouba-kiki

**Author for correspondence:**
Manuel Bohn
e-mail: manuel_bohn@eva.mpg.de

# Bo-NO-bouba-kiki: picture-word mapping but no spontaneous sound symbolic speech-shape mapping in a language trained bonobo

Konstantina Margiotoudi[1,2,3], Manuel Bohn[4], Natalie Schwob[5], Jared Taglialatela[6,7], Friedemann Pulvermüller[1,2,8,9], Amanda Epping[6], Ken Schweller[6] and Matthias Allritz[10]

[1]Brain Language Laboratory, Department of Philosophy and Humanities, WE4, Freie Universität Berlin, Berlin, Germany
[2]Berlin School of Mind and Brain, Humboldt Universität Berlin, Berlin, Germany
[3]Laboratory of Cognitive Psychology, CNRS and Aix-Marseille University, Marseille, France
[4]Department of Comparative Cultural Psychology, Max Planck Institute for Evolutionary Anthropology, Leipzig, Germany
[5]Department of Psychology, The Pennsylvania State University, University Park, PA, USA
[6]Ape Cognition and Conservation Initiative, Des Moines, IA, USA
[7]Department of Ecology, Evolution and Organismal Biology, Kennesaw State University, Kennesaw, GA, USA
[8]Einstein Center for Neurosciences Berlin, Berlin, Germany
[9]Cluster of Excellence 'Matters of Activity', Humboldt-Universität zu Berlin, Berlin, Germany
[10]School of Psychology and Neuroscience, University of St Andrews, St Andrews, Fife KY16 9JP, UK

KM, 0000-0002-9505-3978; MB, 0000-0001-6006-1348; NS, 0000-0001-6151-9568

Humans share the ability to intuitively map 'sharp' or 'round' pseudowords, such as 'bouba' versus 'kiki', to abstract edgy versus round shapes, respectively. This effect, known as sound symbolism, appears early in human development. The phylogenetic origin of this phenomenon, however, is unclear: are humans the only species capable of experiencing correspondences between speech sounds and shapes, or could similar effects be observed in other animals? Thus far, evidence from an *implicit* matching experiment failed to find evidence of this sound symbolic matching in great apes, suggesting its human uniqueness. However, *explicit* tests of sound symbolism have never been conducted with nonhuman great apes. In the present study, a language-competent bonobo completed a cross-modal matching-to-sample task in which he was asked to match spoken English words to pictures, as well as 'sharp' or 'round' pseudowords to shapes. Sound symbolic trials were interspersed among English words. The bonobo matched English words to pictures with high accuracy, but did not show any evidence of spontaneous sound symbolic matching. Our results suggest that speech exposure/comprehension alone cannot explain sound symbolism. This lends plausibility to the hypothesis that biological differences between human and nonhuman primates could account for the putative human specificity of this effect.

## 1. Introduction

In classic semantic theories, the arbitrariness of the linguistic form is a design feature of human language [1,2]. The selection of any linguistic form is arbitrary without any link to its meaning. However, there are instances in human languages where an arbitrary relationship between linguistic form and meaning does not hold. In these cases of sound symbolism, meaningless speech sounds can evoke the meaning of a range of sensory properties [3]. For example, the

immanent mapping between back vowels and large objects, and between front vowels and small objects, known as the /mil/-/mal/ effect, is one of the first descriptions of sound symbolic mappings [4].

Perhaps the most popular demonstration of sound symbolism is the 'maluma-takete' effect described by Köhler [5]. In the 'maluma-takete' example, the 'round' sounding pseudoword 'maluma' fits better to describe an abstract round figure, whereas the 'sharp' sounding pseudoword 'takete' fits better to an abstract edgy figure. In this intuitive sound-shape mapping, combinations of certain speech sounds fit better to express the visual property of a round or of an edgy shape. For example, back vowels versus front vowels [6] or sonorants versus obstruents [7] are better mapped to a round versus an edgy abstract shape, respectively.

Interestingly, the 'maluma-takete' mapping, and later on known as the 'bouba-kiki' mapping [8], has been reported in native speakers of different languages and cultures. For example, native speakers of French [9], English [10], Japanese [11] and Spanish [12] could detect the congruencies between the 'round' sounding pseudoword 'maluma' and a round shape, as well as between the 'sharp' sounding pseudoword 'takete' and an edgy shape. In other words, these different speakers shared the immanent mapping between a sequence of meaningless speech sounds and abstract shapes.

Most importantly, sensitivity in detecting sound symbolic mappings of the 'maluma-takete' type has been reported as early in life as four months of age [13]. A meta-analysis of 11 studies on sound symbolic congruency detection in early development (i.e. 4–38 months) showed that sensitivity to sound-shape mappings is present, but with a moderate effect at a very young age. Notably, this sensitivity is observed first for 'round' sounding pseudowords, followed by 'sharp' ones [14]. The findings on sound symbolic sensitivity early in ontogeny are explained by Fort et al. [14] as an interplay between an innate biological perceptual ability for inferring speech sound-shape associations, and learned sound symbolic regularities in the environment of the child. Nevertheless, there is little evidence suggesting the systematic usage of speech sounds in human languages referring to round or edgy shapes [15,16]. In other words, a scenario in which these sound-shape mappings are learned owing to their systematic presence in the human vocabulary is not well supported.

From an evolutionary perspective, sound symbolic mappings of words and shapes may have played an important role in the initial emergence of protowords and language. Köhler in 1929 [5], when describing the 'maluma-takete' mappings, suggested that the associations between speech sounds and abstract shapes could have been present in early languages. Sound-shape mappings, as a demonstration of non-arbitrariness in human language, could have assisted referential insight in humans [17]. Human ancestors could have understood that certain speech sounds can evoke a specific sensory property, and hence they could more easily map an auditory signal to a specific meaning [18]. As sound symbolic mappings are present across human populations, it seems conceivable that sound-shape congruencies played a role in the formation of protowords. According to this view, humans universally share the immanent mappings that certain speech sounds express more intuitively certain sensory meanings. This shared ability may have provided a background for the emergence of the first linguistic forms

expressing meanings about a range of sensorimotor experiences [19]. Support for this view comes from research in a different modality: gesture. The analogue to sound symbolic words in this domain are iconic gestures—gestures that visually resemble their referent. Laboratory-based experiments and observations of newly emerging sign languages have found that children and adults use iconic gestures to establish communication in the absence of a shared language. With time, these ad hoc signs may transform into language-like systems [20–22].

The discussion surrounding the role of sound symbolism in shaping protolanguages gives rise to the question of the phylogenetic origin of this mapping. Is sound symbolism a human specific ability? Specifically, could our closest relatives, namely nonhuman great apes (hereafter great apes), detect congruencies between speech sounds and abstract visual shapes? Previously, Margiotoudi et al. [23] tested humans as well as a group of touch-screen trained great apes (gorillas and chimpanzees) with a two-alternative forced choice implicit task. In this task, a 'sharp' or 'round' sounding pseudoword preceded the presentation of two abstract shapes (i.e. an edgy and a round shape). The subjects had to select one of the two shapes, in order to pass to the next trial. Neither the great apes nor the humans were explicitly instructed to select the shape that best matched the pseudoword. The results of the two experiments revealed sensitivity to sound symbolic mappings of human speech sounds to shapes only in humans but not in great apes. Humans selected a round shape after the presentation of a 'round' sounding pseudoword, and an edgy shape after a 'sharp' pseudoword significantly more often than chance. Notably, when a second group of human subjects were tested in the same forced choice task but with explicit instructions—namely they were explicitly instructed before the experiment to select the shape that best matched the preceding pseudoword—human participants detected sound symbolic congruencies 10% more often than participants in the implicit experiment. These results suggest that, in contrast to human subjects, great apes do not (or cannot) detect/infer sound symbolic congruencies between meaningless speech sounds and abstract shapes. In summary, human performance on the sound symbolic task, regardless of the given instructions (explicit versus implicit), was significantly above chance. This performance was not detected in the great apes.

These findings suggest that only humans make sound symbolic mappings of 'maluma-takete' type when tested with a forced choice task. However, the differences reported between humans and great apes may also be explained by a mere lack of speech exposure and understanding. Humans are not only exposed to speech stimuli much more than the great apes tested in previous studies, they also learn through this linguistic training that speech sounds can be used to refer to things. This ability might be essential for mapping sound symbolic speech sounds to shapes.

The present study aimed to determine if a language-competent bonobo preferentially selects abstract shapes that are sound-symbolically congruent with meaningless speech sounds in a sound-picture matching task that was as 'explicit' as possible. In order to investigate this question, we tested Kanzi, a language-competent bonobo (Pan paniscus). Kanzi is able to match English words to pictures in a match-to-sample task [24,25]. Because Kanzi performs with high

accuracy in this task (~80% correct) and because he often performs well with novel pictures of known objects, it can be assumed that he follows a strategy of trying to pick from the pictures on the screen the one that best matches the preceding word. Our aim was to use this strategy and to see if Kanzi would apply it in trials with novel stimuli in a way that suggests spontaneous sound-symbolic matching, specifically the bouba-kiki effect. To be clear, the goal was not to train sound-symbolic responding with one set of stimuli and to see if this generalizes to another context, rather, we aimed to test whether Kanzi would intuitively choose sound-symbolic matches when presented with seemingly arbitrary stimuli, as humans do, as long as the task was embedded in a familiar sound-to-picture matching task. Consequently, we presented a set of test trials that allow for sound symbolic matching, embedded in a large number of regular word-to-picture matching trials with which Kanzi is very familiar. By applying the aforementioned design, we were confident that Kanzi would treat the sound symbolic trials similarly to the English word-picture matching trials. In other words, Kanzi would search for congruency between auditory and visual stimuli in English word-picture as well as sound-symbolic trials.

## 2. Design and procedure

The design, procedure and analysis plan were pre-registered at https://osf.io/749pg. The electronic supplementary material, video SVI, experimental stimuli along with the data and the analysis scripts are publicly available in the following repository: https://github.com/manuelbohn/bonoboubakiki.

### (a) Subject

Kanzi is a 40-year old male bonobo who currently lives at the Ape Initiative in Des Moines, Iowa, USA. Kanzi was born at the Yerkes National Primate Research Centerat Emory University, Atlanta, GA, in 1980 where he lived for the first years of his life before moving to Georgia State University's Language Research Center, and then finally to the Ape Initiative (formerly Great Ape Trust). Kanzi was reared by his adopted bonobo mother Matata, and grew up with other bonobos, but also had frequent contact with humans throughout his life. Kanzi's rearing history and descriptions of the language studies he participated in can be found in Savage-Rumbaugh *et al*. [26] and Hillix & Rumbaugh [27]. A recent overview can be found in Krause and Beran [28]. Kanzi has a long history of language exposure, starting from a very young age. He was taught to communicate with humans via a 'lexigram' board, a large board with 395 different symbols or 'lexigrams'. Each lexigram corresponds to an English word and contains an abstract symbol representing that word. Kanzi's lexigram communication with carers includes requests for objects (apple) and activities (play', 'groom), as well as answering questions asked by humans in English (e.g. 'Where would you like to go?'). Kanzi has also demonstrated his spoken language and lexigram competence in cross-modal match-to-sample tasks, matching novel pictures to spoken words [25].

The Ape Initiative is a non-profit bonobo research facility housing seven bonobos who are offered a variety of enrichment activities and research daily. Food and water are never withheld and all participation in research is voluntary and rewarded with additional food. Signs of stress are monitored by care and research staff at all times. Kanzi has a history of participating in research tasks and is housed in accordance with guidelines provided by the USDA and the Association of Zoos & Aquariums. Experimental procedures conformed to the regulations of the Institutional Animal Care and Use Committee of the Ape Initiative, approval number 170904-01R.

### (b) Apparatus

Data were collected on mounted touchscreens in one of two testing rooms (15 sessions each) at Ape Initiative. Both touchscreens were Elo Touch touchscreens (24″ in East Testing Room; 32″ in West Testing Room) with a refresh rate of 60 Hz and resolutions of 1920 by 1080 pixels. Audio stimuli were presented via two Realtek high definition speakers that were either 1 m to the right (East Testing Room) or the left (West Testing Room) of Kanzi. The experimental procedure was written in Java by Ken Schweller and run on a HP Pavilion Laptop. Food rewards were delivered manually through a tube connecting researcher and ape areas (see the electronic supplementary material, SV1).

### (c) Stimuli

The experiment presented two categories of trials in a cross-modal matching-to-sample task. Regular trials consisted of a pre-recorded spoken English word, followed by black and white object photographs as match and foil pictures. Test trials presented pseudowords, which had 'round' or 'sharp' sounding phonetic properties, as audio samples, followed by a presentation of two abstract drawings which contained white blotches with round or edgy outlines on a black background as a hypothetical match and foil.

#### (i) Regular words

One hundred English words were included that referred to animate and inanimate concepts (e.g. juice, bug, onion) Kanzi had previously demonstrated familiarity with [25]. To match the testing words (see below), regular words were recorded in AUDACITY by the same female native Greek speaker who had recorded the sound symbolic pseudowords, which were previously used and validated to elicit sound-shape associations in humans [23]. Word duration ranged from 364 ms to 1216 ms, with a mean duration of $M = 710.85$ (s.d. = 168.22 ms). The recordings were saved at 44.1 kHz sampling rate. All words were normalized for amplitude. All audio stimuli can be found in the electronic supplementary material.

#### (ii) Regular pictures

Using a Google picture search, two different images were found and used as tokens for each English audio recording (200 regular pictures total). All images were edited using ADOBE PHOTOSHOP CS5.1 (Adobe Systems Incorporated, San Jose, CA, USA) to contain a black and white image against a black rectangular background to match the test trial pictures. All stimuli were displayed on a black screen background (figure 1). While Kanzi does not have any experience with the test shapes (see below), he does not necessarily have real-life experience with some of the regular trial images either (e.g. gorilla, phone, balloon). Original pictures ranged

Proc. R. Soc. B 289: 20211717

**Figure 1.** Trial procedure on all trials. For details, see text. (Online version in colour.)

in size between 132 and 4272 by 157–2848 pixels and were automatically scaled by the MTS program to a maximum of 558 width by 294 height while still retaining their original width to height ratio. Images were presented in one of nine screen position grid cells (see below) and each picture served as a match and a foil on different trials.

### (iii) Test pseudowords

Twenty bisyllabic pseudowords (10 'round' and 10 'sharp' sounding) with a consonant-vowel-consonant-vowel (CVCV) structure were included in the experiment. The two selected vowels and consonants in each word were the same, following a CiViCiVi structure (e.g. lolo, kiki; for the list of pseudowords, see [23]). Pseudoword duration ranged from 517 ms to 659 ms, with a mean duration of $M = 578$ (s.d. = 41.28 ms). An example of a 'round' sounding pseudoword of this structure is 'momo', an example of a 'sharp' sounding pseudoword is 'kiki'. All pseudowords were recorded with AUDACITY by a female native Greek speaker in a sound-proof room (2.0.3) and saved at 44.1 kHz sampling rate. Pseudoword sound stimuli had been rated by a multinational sample of human study participants for their perceived roundness versus sharpness prior

to this study, and have been used in a forced choice sound symbolic paradigm conducted with great apes as well as humans [23,29].

### (iv) Test shapes

Twenty white shapes (10 round and 10 edgy) were included in the experiment. Each original test shape was $350 \times 350$ pixels in size and appeared rescaled on the touchscreen at a size of $294 \times 294$ pixels in one of nine possible locations (see below). All shapes were previously rated for perceived roundness versus edginess, and have been used in forced choice sound symbolism tests (see [23,29]) The full list of picture stimuli can be found in the openly accessible online electronic supplementary material of Margiotoudi et al. [23].

### (d) Trial procedure

The general procedure was identical for all trials (figure 1). Stimuli were presented on a black screen. Kanzi initiated each trial by pressing a centralized green circle, which then disappeared. Following a wait interval of 500 ms, an audio recording of an English word (regular trials) or pseudoword (test trials) was played through the speakers. A match and a foil picture appeared in two of nine possible locations on the screen 100 ms after the end of the audio stimulus. Images were either two pictures of objects (regular trials) or two abstract shapes (test trials). As picture positions were chosen randomly, matches and foils appeared with approximately equal frequencies in each of the nine grid positions, ensuring that handedness or location preferences could not bias Kanzi's choice accuracy (regular trials: matches appeared with frequencies ranging from 0.100 to 0.121 in each of the nine positions, foils with frequencies ranging from 0.103 to 0.125; test trials: match position frequencies ranging from 0.093 to 0.153, foil position frequencies ranging from 0.083 to 0.147 across the nine grid positions). Selecting one of the two pictures was followed by a black feedback slide which lasted 2000 ms. On regular trials, if Kanzi gave an incorrect response, this was followed by a 'buzz' sound which is familiar to Kanzi and reliably associated with following incorrect responses. If instead Kanzi gave the correct response, this was followed in 34.5% of cases by a familiar 'chime' sound, and in 65.5% of cases by a 'tadaa' sound and the delivery of a food reward. On test trials, regardless of the 'accuracy' of the sound-image matching, responses were always followed by a 'tick' sound and no food reward. This non-differential reinforcement procedure was chosen to prevent Kanzi from learning about individual shapes, or sound-shape combinations, from feedback. For an example of Kanzi working on this task, see the electronic supplementary material, video SV1.

### (e) Experimental design

Kanzi completed 3000 trials (2700 regular trials and 300 test trials) across a total of 30 daily testing sessions. Each session presented Kanzi with a total of 100 trials, which included 90 regular and 10 test trials. To maximize the possibility that Kanzi would approach test trials with an understanding that he was asked to match the correct image to the audio sample, test trials were distributed among regular trials. As mentioned above, Kanzi has been previously shown to select a correct referent after hearing a word using a two-alternative forced choice task [24,25], which is also a very

common paradigm to test sound shape associations in humans (see [9,10,23], etc.). By embedding test trials within regular word-to-image trials Kanzi has experience with, we hoped to make it clear to Kanzi that what was asked of him was to match an auditory stimulus to one of the two available referents, thus making our task as explicit as possible to a non-linguistic individual. Testing trials were distributed among regular trials in the following manner. Every session was divided into 10 blocks of 10 trials. Each block included nine regular trials and one sound symbolic test trial. The sound symbolic trial occurred at a randomly determined position in each block of 10 trials, but never as the first or last in the block of 10 trials. This ensured that sessions never began or ended with a test trial and that there were never two consecutive test trials.

Across the 2700 regular trials, the 100 English words were sampled between 26 and 28 times, though never more than four times among the 90 regular trials of a single session. Each of the two object pictures that corresponded to the same object served as the match for the corresponding sound sample either 13 or 14 times across all 2700 trials. The accompanying foil picture on a given trial was randomly chosen from the remaining 198 object pictures, counterbalanced across trials such that each of the 200 object pictures was used either 13 or 14 times as a foil picture across all regular trials.[1]

Across the 300 test trials, each of the 20 pseudowords were presented exactly 15 times and were distributed such that there were always five 'round' sound samples and five 'sharp' sound samples presented among the 10 test trials of a single session. The same pseudoword sound sample never occurred more than once in a single session. Each edgy shape served between 8 and 19 times across all test trials as the hypothetical match for a 'sharp' sound sample, and between 11 and 18 times as the hypothetical foil for a 'round' sound sample. Conversely, each round shape served between 12 and 19 times as the hypothetical match for a 'round' sound sample, and between 11 and 22 times as the hypothetical foil for a 'sharp' sound sample.

## 3. Data analysis

We analysed the data using Bayesian generalized linear mixed models. Models were fit in R (R v. 4.0.3; [30]) using the function brm from the package brms [31]. All models included random intercepts for sound and random slopes for trial[2]; model notation: (trial | sound). We used default priors for all parameters. Inference was based on computing a 95% credible interval (CrI) for the posterior distribution of the predictor in question, and checking if it overlapped with 0. A power analysis, as well as a prior sensitivity analysis can be found in the electronic supplementary material, figures S1 and S6.

## 4. Results

Our main question was if Kanzi selected the correct object above a level expected by chance in the sound symbolic test trials. To test this, we fitted an intercept only model to the test trials (correct ∼ 1 + (trial | sound)). The estimate for the intercept represents the average rate of correct responses in link space; with an intercept of 0 corresponding to a

performance at chance level (50% correct). We found that the 95% CrI for the intercept included 0 ($\beta = 0.16$, 95% CrI = [−0.09–0.41]); thus, Kanzi did not reliably select the correct shape above chance (figure 2a). His performance fluctuated around chance level throughout the experiment, with no clear signs of an in- or decrease in performance over time (figure 2c).

By contrast, when we fitted the same model to the regular trials, we found that Kanzi selected the correct picture well above chance ($\beta = 1.99$, 95% CrI = [1.69–2.31]). After hearing an English word, he selected the correct picture at an astonishing rate of 87% according to the model (figure 2a). Moreover, his performance was consistently high across test sessions (figure 2c).

When visualizing reaction times for both regular and test trials, we found that Kanzi's reaction times were somewhat faster in test trials compared to regular trials, and that the distribution for response times in regular trials was wider, both for correct and incorrect trials (figure 2b). This might reflect a more deliberate choice process in regular trials, a point that will be further discussed below. A more detailed exploratory comparison revealed no difference in the extent to which response times decreased across regular versus test trials (see the electronic supplementary material, figure S3).

Our secondary question was whether Kanzi was more likely to select the correct shape in sound symbolic test trials when the sound corresponded to a round as opposed to an edgy shape. We fitted a model predicting correct choices by the shape of the target (correct∼target_shape + (trial | sound)). The estimate for target shape was not reliably different from 0 ($\beta = 0.17$, 95% CrI = [−0.35–0.68]), suggesting that Kanzi's performance was not systematically influenced by the target's shape. Figure 2d visualizes the results. A breakdown of Kanzi's performance for individual stimuli can be found in the electronic supplementary material, figure S10.

## 5. Discussion

In the present study, we tested a language-competent bonobo in an explicit match-to-sample task on sound symbolic mappings of the 'maluma-takete', or 'bouba-kiki' type. Much like human tests of sound shape associations, every sound symbolic trial included the presentation of a 'round' or 'sharp' pseudoword, followed by two abstract shapes. By introducing sound symbolic trials in a previously learned match-to-sample task including familiar stimuli edited to appear more similar to the abstract shape stimuli, we anticipated that Kanzi would treat the sound symbolic trials similarly to the familiar trials, namely selecting the picture that matched the presented sound.

Kanzi correctly matched spoken English words to their corresponding pictures at a rate significantly above chance. However, Kanzi did not select a round shape when a 'round' sounding pseudoword was presented, nor an edgy shape after the presentation of a 'sharp' sounding pseudoword. The present findings show that for a language-competent ape who was asked, with instructions that are as explicit as possible, to match a pseudoword to an abstract shape, no spontaneous sound symbolic mappings could be detected.

The present results are consistent with the findings of Margiotoudi *et al.* [23]. In their study, no significant sound

Proc. R. Soc. B 289: 20211717

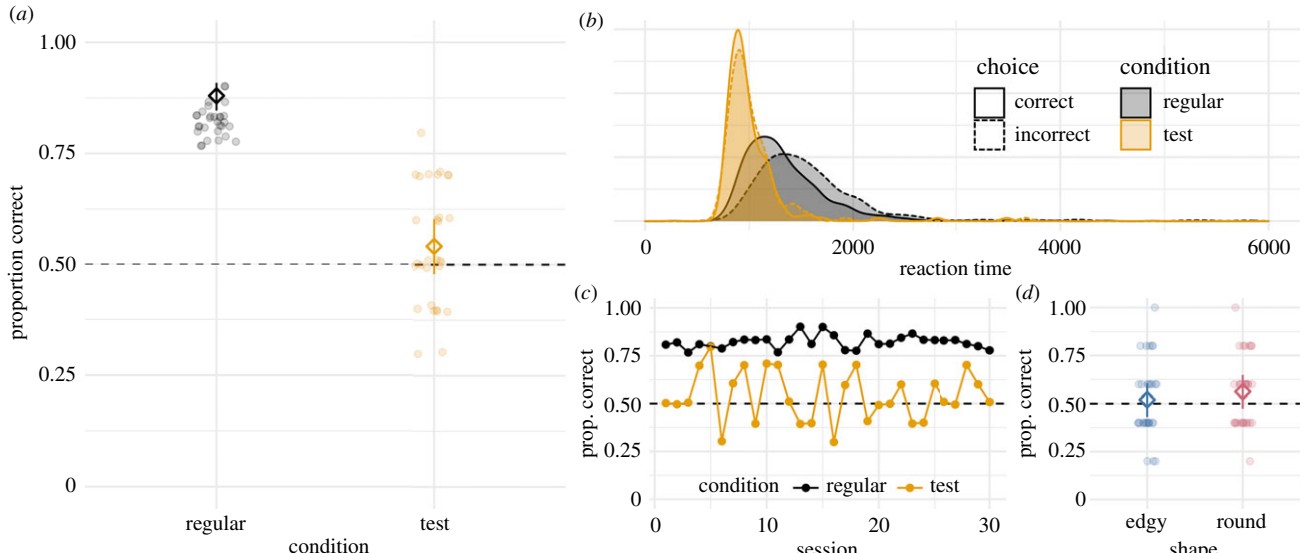

**Figure 2.** (a) Proportion of correct choices in regular and test trials. (b) Distribution of reaction times in regular and test trials for correct and incorrect choices. (c) Proportion of correct choices in each session for regular and test trials. (d) Proportion of correct choices in test trials with either edgy or round target shapes. In (a,d) diamonds and error bars represent the mean and 95% CrI based on the models' posterior distribution. Transparent dots show horizontally jittered session means of the data. (Online version in colour.)

symbolic congruent matching performance was observed in a group of great apes, when tested with an implicit two-alternative forced choice task. By contrast to the great apes tested in Margiotoudi *et al.* [23], Kanzi has been exposed to, and comprehends, a significant number of English spoken words (*ca* 200). Nevertheless, the unique rearing experience of Kanzi in a human language environment did not result in him being also sensitive to sound symbolic matching.

One factor that could have conferred an advantage to Kanzi in sound symbolic congruency detection, had he shown it, is prior exposure to similar mappings in familiar words, an exposure that the subjects in the study by Margiotoudi *et al.* [23] did not have. If the vocabulary familiar to Kanzi was to a certain degree iconic or onomatopoeic—that is, the linguistic form resembled its meaning [18]—then one might expect that Kanzi may be able to detect novel mappings between meaningless speech sounds and abstract visual shapes. However, none of the words familiar to Kanzi was onomatopoeic—namely the linguistic form did not resemble another sound (e.g. an animal sound), nor referred to tactile or motor experiences which have been also linked to vocal iconicity [32]. Hence Kanzi's vocabulary had no signs of non-arbitrary links between linguistic forms and meanings. Nevertheless, some words familiar to Kanzi refer to round or sharp objects (e.g. ball, knife). A recent study by Sidhu *et al.* [16] showed that 'maluma-takete' mappings are present in the English vocabulary. Specifically, words related to round objects are characterized by the 'round' sounding phonemes (e.g. /u/, /b/, /m/), whereas words referring to spiky objects included 'sharp' sounding phonemes (e.g. /k/, /t/, /i/). Considering that Kanzi has been exposed to some of these words (e.g. ball), it appears that the sound symbolism that is present in the English vocabulary was not sufficient to affect Kanzi's performance on the 'bouba-kiki' effect. Given that the findings regarding the innateness of sound symbolic mappings in humans are also not conclusive [14], we may conclude that exposure to sound symbolic mappings might not be a determining factor in detecting sound symbolic mappings.

Another hypothesis for how human speech exposure could hypothetically have facilitated sound symbolic congruency detection in Kanzi would suggest that Kanzi is more habituated to human speech and to the phonemic properties of speech signals. Consequently, Kanzi could more easily differentiate 'sharp'/'round' sounding pseudowords, compared to other non-language competent great apes. However, prior research generally confirms that some nonhuman primates can also differentiate human speech sounds regardless of their language competence [33,34]. For that reason, we think it is implausible that Kanzi had an advantage over other great apes when it comes to differentiating the phonemic speech properties of 'round' versus 'sharp' pseudowords.

Kanzi showed no sensitivity in detecting congruencies between pseudowords and shapes and therefore no advantage compared to other non-language trained great apes who were tested on a sound symbolic task [23]. Considered together, the findings on sound symbolic congruency detection from both studies support the hypothesis that this sound symbolic ability is unique to humans.

The absence of sound-symbolic matching of the 'maluma-takete' type in the present study echoes the results found in research on great apes's comprehension of iconic gestures. When a human experimenter used a novel iconic gesture to refer to an apparatus, great apes failed to use this cue [35,36]. Adding a communicative training and an iconic vocalization to the gesture also did not enhance performance [37]. Negative results were also found when the gesture was used to instruct great apes in how to open an apparatus [38]. However, these studies did not include language-competent apes like Kanzi and so it remains unclear if language training has an effect on iconic signal comprehension in the gestural modality.

One limitation of our study is the qualitative difference between stimuli in regular trials and test trials. Though a lot of efforts were taken to make regular trials and test trials homogeneous (using black and white images in all trials, using an identical speaker for all sound presentations), critical differences remained between the stimuli in regular

trials (familiar English words paired with pictures of known real world objects) and the stimuli in test trials (unfamiliar pseudowords with no previously learned relationship to the subsequently presented abstract shapes). This is a direct consequence of our experimental aims: in order to test whether Kanzi would spontaneously show sound-symbolic mappings when presented with stimuli that had already been validated with human subjects, we embedded these same stimuli in a task that was as familiar as possible to Kanzi. In addition, the reward schedules differed: regular trials were differentially reinforced (food reward for correct choice in some trials) to keep Kanzi motivated to participate and to create an expectation that all trials have a hypothetical correct answer. Performance in test trials, on the other hand, was never reinforced, irrespective of choice, to avoid as well as possible that Kanzi could learn about sound-shape correspondences from feedback. These differences were introduced to maximize comparability between human performance in the boubakiki task and Kanzi's spontaneous behaviour. However, they also create a trade-off: one might argue that because the stimuli in test trials were so different, and completely unfamiliar to Kanzi, that perhaps a response strategy of identifying the picture that best matches the word just heard did not fully carry over from regular to test trials. Indeed, though Kanzi's responses did not become faster to a greater extent across test trials than they did in regular trials (see the electronic supplementary material, figure S3), the different response time distributions for test versus regular trials (figure 2b) may indicate that Kanzi responded to these trials with different strategies. While this is not evidence that Kanzi's strategy in test trials was *not* to find the best match for the sample, future studies may still aim to further increase the likelihood of carrying over the task instruction (find the picture that matches the word) by adding trials that share characteristics of both the regular trials used here and the test trials, e.g. novel English word trials paired with novel pictures, including drawings, that are presented repeatedly and have a clear, always rewarded solution.

Finally, while our results presented, to our knowledge, the most explicit version of a sound-symbolic matching task to a nonhuman ape to date, and did not find spontaneous mapping, the ability to make such mappings may still be detected with alternative experimental paradigms. The present study aimed to test the intuitive mappings between meaningless speech sounds and shapes in a language-competent ape within a match-to-sample task. It did not investigate the question of whether there are any differences, e.g. in the *speed of learning* after repeated exposure of arbitrary audiovisual mappings compared to the speed of learning of sound symbolic mappings. Future studies should thus explore further whether nonhuman apes can learn faster sound symbolic mappings between, e.g. 'round' and 'sharp' pseudowords and shapes, as well as other sound symbolic mappings of audiovisual features (i.e. speech sounds and size of objects). Interestingly, Bohn and colleagues found such a pattern for iconic gestures [35], that is chimpanzees were faster to learn that an iconic gesture identified a specific apparatus as compared to an arbitrary gesture.

Apart from the 'maluma-takete' mapping, there are other sound symbolic mappings to which non-human primates might be sensitive, because they could be more relevant to their experiences. A worthwhile mapping to study is the 'mil-mal' effect, in which the high front vowel /i/ fits better

to describe something small, whereas the vowel /a/ fits better to describe a large referent [4]. In accordance with the frequency code theory [39], humans match low (high) frequencies, such as vowels with low (high) second formant to large (small) sizes owing to the statistical co-occurrences of these audiovisual properties in our natural environment [40,41]. Non-human primates could show sensitivity in matching low (high) frequencies to large (small) sizes, because: (i) they are plausibly exposed to these statistical co-occurrences in their natural environment, and (ii) have the necessary neuroanatomical infrastructure for carrying the learning of these statistical regularities. Specifically, great apes show an expanded occipito-temporal white matter tract connectivity similar to humans that could permit the associative learning between these 'low-level' audiovisual features of pitch and size [42]. The frequency-size mapping and other sound symbolic mappings need to be tested in non-human primates in order to acquire a comprehensive view on the evolutionary continuity of sound symbolic phenomena.

The absence of sensitivity to sound symbolic mappings of the 'maluma-takete' type by Kanzi and other great apes is consistent with one recent proposal for the mechanism that may underlie sound symbolic mappings of the same type in humans [29]. In their study, human participants performed a classic forced choice task for a sound symbolic condition and for a second condition in which they had to match 'round' and 'sharp' action sounds to round or edgy action shapes. These action sounds and shapes were previously recorded from different hand drawings on a piece of paper. Performance in both forced choice conditions (sound symbolism and action sound-shape) was significantly above chance and, critically, correlated with each other. Good sound symbolic mappers were also good action sound-shape mappers. Based on these results, the authors suggested that sound symbolism may be related to action knowledge, and could possibly be a by-product of this hand action knowledge. Comparing the visual and auditory features of the 'maluma-takete' mappings to the visual and auditory outputs of hand actions, one can detect physical similarities between the two domains. For instance, a sharp hand movement while drawing on a piece of paper results in a visual output that resembles the abstract spiky 'takete' figure. Similarly, the sound produced by this sharp hand movement exhibits acoustic similarities to a 'sharp' sounding pseudoword, such as 'takete'. Both sounds—the 'takete' pseudoword and the 'sharp' action sound—are characterized by abrupt/sudden transitions in their frequencies and more power in higher frequencies compared to a 'round' pseudoword or a round action sound. The visual and auditory similarities between hand action sounds and shapes, and between sound symbolic pseudowords and abstract shapes, were observed for the round stimuli as well.

In order for the memories of hand action knowledge to be grounded in sensorimotor systems, distributed neuronal networks are required, which would link modality-specific and multimodal areas in the brain's cortex (see [43] for a review). Two important prerequisites for the emergence of such distributed neuronal circuits are: (i) the association between the motor, visual and acoustic products of our hand action knowledge under biological associative Hebbian learning [44], and (ii) strong long cortical connections that would permit links between motor and sensory cortices in the brain. Comparative neuroanatomical evidence has shown that

nonhuman primates do not have a human-like neuroanatomical infrastructure of long white matter tracts in their brains, relevant to sensorimotor integration. Specifically, long white matter cortical connections between frontal and posterior temporal cortices, such as the arcuate fasciculus [45,46] or the superior longitudinal fasciculus [47] are not as strongly developed in great apes. If the hand action knowledge hypothesis for sound symbolism proposed by Margiotoudi & Pulvermüller [29] has merit, these neuroanatomical differences between humans and great apes could potentially account for the differences between species in sound symbolic performance. In this view, sound symbolism, as a by-product of the physical properties of the multimodal features of hand actions, would recruit the same distributed neuronal circuits that carry the knowledge of hand actions, and would hence require strong connectivity between frontal and temporal areas in the cortex.

In order to test the proposals that: (i) sound symbolic mapping of maluma-takete-like pseudowords and abstract round versus edgy shapes is a by-product of knowledge about the acoustic-visual aspects of shapes and sounds elicited by hand actions performed with tools [29]; and that (ii) an intact human frontotemporal anatomical connection and in particular arcuate fasciculus is necessary for using this transmodal knowledge, future studies would be desirable. For further exploring the first proposal, it would be essential to investigate infants of about four months of age or younger who do not yet show consistent sound symbolic behavioural effects [14] but already have an intact arcuate fasciculus [48]. If these individuals show sound symbolism after learning hand/tool action related congruencies, this could be regarded as favourable evidence. A further study with Kanzi and other great apes could address the second proposal. If nonhuman great apes also failed to learn the transmodal association between action sounds and the visual traces of motor trajectories in the absence of sound symbolic knowledge, there would be further support; success on sound symbolic tasks after successful learning of transmodal visual and auditory action associations could be regarded as evidence counter to the proposal.

## 6. Summary

The present findings indicate that when a language-competent bonobo was tested on sound symbolic mappings under an explicit match-to-sample task, he did not detect congruencies between 'sharp' (round) meaningless speech sounds and edgy (round) abstract shapes. These findings support the hypothesis that sound symbolism is an ability specific to humans. From an evolutionary perspective, the human specificity of sound symbolism might be explained by neuroanatomical differences between human and nonhuman primates.

Ethics. Experimental procedures conformed to the regulations of the Institutional Animal Care and Use Committee of the Ape Initiative, approval number 170904-01R.

Data accessibility. Data can be found at: https://github.com/manuelbohn/bonoboubakiki.

Authors' contributions. K.M.: conceptualization, methodology, project administration, writing—original draft; M.B.: conceptualization, formal analysis, methodology, writing—original draft; N.S.: conceptualization, formal analysis, methodology, writing—original draft; J.T.: conceptualization, methodology, writing—original draft; F.P.: conceptualization, methodology, writing—original draft; A.E.: methodology; K.S.: methodology; M.A.: conceptualization, investigation, methodology, formal analysis, project administration, writing—original draft. All authors gave final approval for publication and agreed to be held accountable for the work performed therein.

Competing interests. We declare we have no competing interests.

Funding. Open access funding provided by the Max Planck Society.

This work was supported by the Deutsche Forschungsgemeinschaft (DFG, German Research Foundation) under Germany's Excellence Strategy through EXC 2025/1 'Matters of Activity (MoA)' and by the 'The Sound of Meaning (SOM)', Pu 97/22–1 and 'Phonological Networks (PhoNet)', Pu 97/25-1. K.M. was supported by the Berlin School of Mind and Brain, by the Onassis foundation, and by the Fyssen foundation. M.A. was supported by the European Research Council under the European Union's Seventh Framework Program (FP7/2007-2013)/ERC grant agreement no. 609819, SOMICS.

Acknowledgements. The authors would like to thank all the animal carers and research staff of the Ape Initiative for their help with the experiment.

## Endnotes

[1]Owing to a software error, there was no audio playback on the first trial of a given session. This problem was solved beginning in session 3 by adding an additional 'filler trial' at the beginning of each session that preceded the presentation of the originally scheduled 100 regular and test trials. The two first trials from sessions 1 and 2 were removed from the dataset, leaving 2698 regular trials and all 300 test trials in the final dataset.

[2]Our pre-registered analysis included random effects for the specific combination of shapes displayed on the screen in each trial. However, it turned out that each combination appeared very infrequently (149 different combinations in 300 test trials; 2612 combinations in 2698 regular trials). We therefore removed this random effect. Including it does not change the results (see analysis script in the online repository).

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
