## [Peer Review File · Proceedings of the Royal Society B: Biological Sciences]

Review History

RSPB-2021-1717.R0 (Original submission)

Review form: Reviewer 1

Recommendation

Major revision is needed (please make suggestions in comments)

Scientific importance: Is the manuscript an original and important contribution to its field?

Good

General interest: Is the paper of sufficient general interest?

Excellent

Quality of the paper: Is the overall quality of the paper suitable?

Acceptable

Is the length of the paper justified?

Yes

Should the paper be seen by a specialist statistical reviewer?

No

Do you have any concerns about statistical analyses in this paper? If so, please specify them explicitly in your report.

Yes

It is a condition of publication that authors make their supporting data, code and materials available - either as supplementary material or hosted in an external repository. Please rate, if applicable, the supporting data on the following criteria.

Is it accessible?

Yes

Is it clear?

Yes

Is it adequate?

Yes

Do you have any ethical concerns with this paper?

No

Comments to the Author

See attached file. (See Appendix A)

Review form: Reviewer 2

Recommendation

Accept with minor revision (please list in comments)

Scientific importance: Is the manuscript an original and important contribution to its field?

Good

General interest: Is the paper of sufficient general interest?

Good

Quality of the paper: Is the overall quality of the paper suitable?

Good

Is the length of the paper justified?

No

Should the paper be seen by a specialist statistical reviewer?

No

Do you have any concerns about statistical analyses in this paper? If so, please specify them explicitly in your report.

No

It is a condition of publication that authors make their supporting data, code and materials available - either as supplementary material or hosted in an external repository. Please rate, if applicable, the supporting data on the following criteria.

Is it accessible?

Yes

Is it clear?

Yes

Is it adequate?

Yes

Do you have any ethical concerns with this paper?

No

Comments to the Author

The study „Bo-NO-bouba-kiki: Picture-word mapping but not sound symbolism in a language trained bonobo“ is in general a well-designed study asking whether the language trained bonobo Kanzi is able to use sound symbolism in a matching-to-sample task. The methodological design and the results are well presented. I have only one major comment.

In total Kanzi completed 3000 trials, 2700 regular trials and 300 test trials. In the 2700 regular trials the task was to match English sounds like apple or knife to the correct picture, independent of any sound symbolism. That means that Kanzi must remember the correct picture of certain sounds. It could be that this sound/ picture pair followed sound symbolism rules or not. But important to solve the task was that Kanzi had this sound/ picture pairs in his memory. In one of ten cases Kanzi get a sound with an unknown picture. My question is: Why should Kanzi come up with the idea that using sound symbolism is a good strategy to solve this task. In case that all known pairs followed a sound symbolism rule this could be a good strategy, but the authors used although a lot of pairs which did not followed such a rule. Therefore, a 50% decision in case Kanzi did not know the sound/ picture pair seems to me the best possible strategy.

Minor comment:

I think it good be helpful for some readers if the authors give a short explanation, why the used Bayesian Generalized Linear Mixed Model.

Decision letter (RSPB-2021-1717.R0)

20-Sep-2021

Dear Dr Margiotoudi:

Your manuscript has now been peer reviewed and the reviews have been assessed by an Associate Editor. The reviewers' comments (not including confidential comments to the Editor) and the comments from the Associate Editor are included at the end of this email for your reference. As you will see, the reviewers and the Editors have raised some concerns with your manuscript and we would like to invite you to revise your manuscript to address them.

To submit your revision please log into <http://mc.manuscriptcentral.com/prsb> and enter your Author Centre, where you will find your manuscript title listed under "Manuscripts with

Decisions." Under "Actions", click on "Create a Revision". Your manuscript number has been appended to denote a revision.

Research ethics:

Use of animals and field studies:

It is a condition of publication that you make available the data and research materials supporting the results in the article. Please see our Data Sharing Policies (<https://royalsociety.org/journals/authors/author-guidelines/#data>). Datasets should be deposited in an appropriate publicly available repository and details of the associated accession number, link or DOI to the datasets must be included in the Data Accessibility section of the article (<https://royalsociety.org/journals/ethics-policies/data-sharing-mining/>). Reference(s) to datasets should also be included in the reference list of the article with DOIs (where available).

All supplementary materials accompanying an accepted article will be treated as in their final form. They will be published alongside the paper on the journal website and posted on the online figshare repository. Files on figshare will be made available approximately one week before the

accompanying article so that the supplementary material can be attributed a unique DOI. Please try to submit all supplementary material as a single file.

Please submit a copy of your revised paper within three weeks. If we do not hear from you within this time your manuscript will be rejected. If you are unable to meet this deadline please let us know as soon as possible, as we may be able to grant a short extension.

Best wishes,
Dr Robert Barton
mailto:proceedingsb@royalsociety.org

Associate Editor
Board Member: 1
Comments to Author:

The authors present a study investigating whether a language trained bonobo is sensitive to sound symbolism. The study is likely to generate broad interest and the authors explain its importance to our understanding of the evolution of human language with clarity. Although both reviewers evaluate the study favorably, they raise several elements that will need to be addressed before the manuscript is acceptable for publication. I agree with reviewer 1 that certain aspects of the task may have obscured the intended referential interpretation of the words that were presented. Similarly related to the nature of the task, reviewer 2 raises the concern that the best strategy to solve the test trials may be a 50% decision. Equally important are reviewer 2's comments concerning the absence of priors in the Bayesian estimation (either appropriate priors should be used, or additional justification should be given why default values were used) and the scope of the interpretation of the results.

Reviewer(s)' Comments to Author:
Referee: 1
Comments to the Author(s)
See attached file

Referee: 2
Comments to the Author(s)

The study „Bo-NO-bouba-kiki: Picture-word mapping but not sound symbolism in a language trained bonobo“ is in general a well-designed study asking whether the language trained bonobo Kanzi is able to use sound symbolism in a matching-to-sample task. The methodological design and the results are well presented. I have only one major comment.

In total Kanzi completed 3000 trials, 2700 regular trials and 300 test trials. In the 2700 regular trials the task was to match English sounds like apple or knife to the correct picture, independent of any sound symbolism. That means that Kanzi must remember the correct picture of certain sounds. It could be that this sound/ picture pair followed sound symbolism rules or not. But important to solve the task was that Kanzi had this sound/ picture pairs in his memory. In one of ten cases Kanzi get a sound with an unknown picture. My question is: Why should Kanzi come up with the idea that using sound symbolism is a good strategy to solve this task. In case that all known pairs followed a sound symbolism rule this could be a good strategy, but the authors used

although a lot of pairs which did not followed such a rule. Therefore, a 50% decision in case Kanzi did not know the sound/ picture pair seems to me the best possible strategy.

Minor comment:

I think it good be helpful for some readers if the authors give a short explanation, why the used Bayesian Generalized Linear Mixed Model.

Author's Response to Decision Letter for (RSPB-2021-1717.R0)

See Appendix B.

RSPB-2021-1717.R1 (Revision)

Review form: Reviewer 1

Recommendation

Accept with minor revision (please list in comments)

Scientific importance: Is the manuscript an original and important contribution to its field?

Good

General interest: Is the paper of sufficient general interest?

Excellent

Quality of the paper: Is the overall quality of the paper suitable?

Good

Is the length of the paper justified?

Yes

Should the paper be seen by a specialist statistical reviewer?

No

Do you have any concerns about statistical analyses in this paper? If so, please specify them explicitly in your report.

No

It is a condition of publication that authors make their supporting data, code and materials available - either as supplementary material or hosted in an external repository. Please rate, if applicable, the supporting data on the following criteria.

Is it accessible?

Yes

Is it clear?

Yes

Is it adequate?

Yes

Do you have any ethical concerns with this paper?

No

Comments to the Author

The authors addressed many of my prior comments in detail. I appreciate the new analyses in the appendix. At the same time, there is still one issue that I think is not addressed sufficiently. In particular, the sound-shape association is one type of sound symbolic association, and not necessarily a representative one in terms of mechanism. Therefore, it is quite possible that Kanzi would not exhibit sound-shape sound symbolism but would exhibit other types of sound symbolism (such as pitch-size). The authors addressed my prior comment by changing the title and in their phrasing in some locations. At the same time, they never explicitly acknowledge the fact that it is possible that had they tested Kanzi with a different sound symbolic association, he would have exhibited sound symbolism. That is, the problem might be due to the fact that this specific association is not one that fits with his experience or that he can represent, but that he would be sensitive to sound symbolism that fits with his experience in the world.

Another more minor issue is that when discussing differences between the two tasks, it is worth mentioning that Kanzi was rewarded for his responses on the regular trials, but not on the test trials. This could have influenced his motivation to respond in the test trial and explain the fast and chance responses. The authors responded that he wasn't rewarded 100% of the time in the regular trials. True, but he was rewarded on the majority of them (67%). Had they thought that rewards would not motivate behavior, reward probably wouldn't have been included at all.

Review form: Reviewer 2**Recommendation**

Accept with minor revision (please list in comments)

Scientific importance: Is the manuscript an original and important contribution to its field?

Good

General interest: Is the paper of sufficient general interest?

Good

Quality of the paper: Is the overall quality of the paper suitable?

Good

Is the length of the paper justified?

Yes

Should the paper be seen by a specialist statistical reviewer?

No

Do you have any concerns about statistical analyses in this paper? If so, please specify them explicitly in your report.

No

It is a condition of publication that authors make their supporting data, code and materials available - either as supplementary material or hosted in an external repository. Please rate, if applicable, the supporting data on the following criteria.

Is it accessible?

N/A

Is it clear?

Yes

Is it adequate?

Yes

Do you have any ethical concerns with this paper?

No

Comments to the Author

The authors made an extensive revision and discuss carefully my concerns. The authors added several paragraphs in the introduction and discussion to discuss the problematic with differences in task demands (regular vs. test trials), and added a number of additional analyses to investigate whether Kanzi approached these trials with different strategies. They specified that their research question, whether Kanzi would offer spontaneously sound-symbolic matchings as "best guess", akin to humans showing sound-symbolic matching intuitively when given the same unfamiliar stimuli.

I think it would nice to find this clarification that Kanzi showed no "spontaneous" sound symbolic matching either in the abstract or title.

Decision letter (RSPB-2021-1717.R1)

02-Dec-2021

Dear Dr Margiotoudi:

Your manuscript has now been peer reviewed and the reviews have been assessed by an Associate Editor. The reviewers' comments (not including confidential comments to the Editor) and the comments from the Associate Editor are included at the end of this email for your reference. As you will see, the reviewers and the Editors have raised some concerns with your manuscript and we would like to invite you to revise your manuscript to address them.

We do not allow multiple rounds of revision so we urge you to make every effort to fully address all of the comments at this stage. Since this is the second round already, this really is the last chance to address remaining concerns. If deemed necessary by the Associate Editor, your manuscript will be sent back to one or more of the original reviewers for assessment. If the original reviewers are not available we may invite new reviewers. Please note that we cannot guarantee eventual acceptance of your manuscript at this stage.

Research ethics:

Use of animals and field studies:

It is a condition of publication that you make available the data and research materials supporting the results in the article (<https://royalsociety.org/journals/authors/author-guidelines/#data>). Datasets should be deposited in an appropriate publicly available repository and details of the associated accession number, link or DOI to the datasets must be included in the Data Accessibility section of the article (<https://royalsociety.org/journals/ethics-policies/data-sharing-mining/>). Reference(s) to datasets should also be included in the reference list of the article with DOIs (where available).

Please submit a copy of your revised paper within three weeks. If we do not hear from you within this time your manuscript will be rejected. If you are unable to meet this deadline please let us know as soon as possible, as we may be able to grant a short extension.

Best wishes,
 Dr Robert Barton
 Editor, Proceedings B
 mailto: proceedingsb@royalsociety.org

Associate Editor
 Board Member: 1

Comments to Author:

The authors completed an in-depth revision in response to the reviewer's comments and I believe this has further improved the quality of the paper. I do agree with both reviewers, however, that some minor, yet crucial nuances should still be clarified before moving towards publication.

Reviewer 1 notes the absence of an explicit acknowledgment that sound symbolism could occur if Kanzi were confronted with a sound symbolic association that would fit with his experience in the world. Reviewer 2 notes that the clarification that Kanzi showed no spontaneous sound symbolic matching should be noted earlier in the manuscript.

Reviewer(s)' Comments to Author:

Referee: 1

Comments to the Author(s)

The authors addressed many of my prior comments in detail. I appreciate the new analyses in the appendix. At the same time, there is still one issue that I think is not addressed sufficiently. In particular, the sound-shape association is one type of sound symbolic association, and not necessarily a representative one in terms of mechanism. Therefore, it is quite possible that Kanzi would not exhibit sound-shape sound symbolism but would exhibit other types of sound symbolism (such as pitch-size). The authors addressed my prior comment by changing the title and in their phrasing in some locations. At the same time, they never explicitly acknowledge the fact that it is possible that had they tested Kanzi with a different sound symbolic association, he would have exhibited sound symbolism. That is, the problem might be due to the fact that this specific association is not one that fits with his experience or that he can represent, but that he would be sensitive to sound symbolism that fits with his experience in the world.

Another more minor issue is that when discussing differences between the two tasks, it is worth mentioning that Kanzi was rewarded for his responses on the regular trials, but not on the test trials. This could have influenced his motivation to respond in the test trial and explain the fast and chance responses. The authors responded that he wasn't rewarded 100% of the time in the regular trials. True, but he was rewarded on the majority of them (67%). Had they thought that rewards would not motivate behavior, reward probably wouldn't have been included at all.

Referee: 2

Comments to the Author(s)

The authors made an extensive revision and discuss carefully my concerns. The authors added several paragraphs in the introduction and discussion to discuss the problematic with differences in task demands (regular vs. test trials), and added a number of additional analyses to investigate whether Kanzi approached these trials with different strategies. They specified that their research question, whether Kanzi would offer spontaneously sound-symbolic matchings as "best guess", akin to humans showing sound-symbolic matching intuitively when given the same unfamiliar stimuli.

I think it would nice to find this clarification that Kanzi showed no "spontaneous" sound symbolic matching either in the abstract or title.

Author's Response to Decision Letter for (RSPB-2021-1717.R1)

See Appendix C.

Decision letter (RSPB-2021-1717.R2)

04-Jan-2022

Dear Dr Margiotoudi

I am pleased to inform you that your manuscript entitled "Bo-NO-bouba-kiki: Picture-word mapping but no spontaneous sound symbolic speech-shape mapping in a language trained bonobo." has been accepted for publication in Proceedings B.

Data Accessibility section

Open Access

Paper charges

Sincerely,
Dr Robert Barton
Editor, Proceedings B
mailto: proceedingsb@royalsociety.org

Associate Editor:
Board Member
Comments to Author:
(There are no comments.)

Appendix A

RE: RSPB-2021-1717

Review of “Bo-NO-bouba-kiki: Picture-word mapping but not sound symbolism in a language trained bonobo”

The authors’ goal is to test whether apes are sensitive to sound symbolism. While they are not the first to test this question, they rightly point out that prior studies might have failed to find such sensitivity because the tested apes did not understand the task, as they were not aware of the potential referential role of sounds. By testing Kanzi, the authors conducted a more informative test of the question. As Kanzi is a unique resource, many would be interested in reading this paper, so it should be published. That said, there are several non-trivial issues with the design which make it difficult to draw clear inferences from the results. The authors should acknowledge and discuss these. There are several additional issues regarding the stimuli, analysis and generalizability of the results to other types of sound symbolism. Some of these can be easily addressed with new analyses. I explain all these points in detail below.

Did Kanzi understand and try to engage with the test trials?

- The main contribution of the paper over prior studies that tested sound symbolism in apes is the use of an ape that understands that words are referential. Certain aspects of the task, however, might have obscured this referential interpretation or demotivated Kanzi from engaging with the test trials as he would with regular trials.
 - o The labels in the test trials were not truly referential, or potentially not referential in a way that Kanzi might understand. That is, whereas in the regular trials each word was associated with a specific object, there was no such relationship between labels and target objects in the test trials. That is, the correct target of a specific word in the regular trials was a specific object, but the correct target of a specific sharp or round word in the test trials was a different shape each time it appeared. Thus, since Kanzi saw the same labels appearing each time with different items, he might have concluded that the sounds are not referential as they don’t function the way that other words that he learned function.
 - o The regular trials used pictures of real world objects. The test trials used abstract drawings. It is unclear whether Kanzi has ever learned to label abstract drawings or understands that words can be referential when there is no real world reference that he can recognize.
 - o Kanzi did not receive any feedback, including not receiving a reward. The authors explain that this was intended to prevent learning. This is understandable (though sound symbolism is often tested with learning paradigms). Nevertheless, as the trials were clearly distinguishable from the others (unfamiliar words, abstract drawings etc.), Kanzi could have quickly realize that there is no point in putting in the effort as he would not be rewarded and would not even know if he were correct.
 - o Indeed, the completely different RT distribution for test and regular trials suggests that Kanzi did not attempt to solve the test trials but just respond quickly and move on. The authors state that shorter RTs do not indicate less effort because, in the regular trials, shorter RTs were associated with more correct responses. The differences are not the same though. The shape of the RT distributions of the correct and incorrect regular trials are identical, only slightly shifted. The RT distribution of the test trials is different, extremely narrow, and shifted to a large degree. It very much looks like a floor effect of responding as quickly as possible.

Stimuli

- The authors used the stimuli from Margiotoudi et al. (2019). While the prior study found an effect with humans with these stimuli, it is not fully clear how the supposedly sound symbolic words were generated. In Margiotoudi et al. (2019) the authors mentioned relying on prior studies, but examination of those studies does not align well with the generated words. For example, Margiotoudi et al. (2019) cite McCormick et al. (2015) as the basis for their phoneme selection, but three of the five phonemes that were used to create sharp words (/s/, /f/, and /z/) were rated as in-between sharp and round in McCormick et al. (2015). In fact, their ratings were very similar to those of /d/, which was used as a round phoneme. It might be better to run an additional analysis of the data, including only words constructed with phonemes that prior studies suggest are sound symbolic, namely, the subset of sharp words with /k/ and /p/ and the subset of round words with /l/, /m/, and /n/.

Analysis

- The authors stated that they used the default values in their Bayesian analysis. This is not recommended. It matters greatly which priors are chosen and they should be theoretically and/or empirically motivated (e.g., consider the effect size in humans and the difference between effects sizes in humans vs apes). For example, selecting too large of an expected effect size will show a null effect even when there is a difference, just not as large as the default setting prescribes. While the plot suggests there really is no effect, the test should be done with appropriate parameters. This will become important if additional analyses are added (e.g., on a subset of the words – see comment above).
- Have the authors conducted any power analysis to determine whether it is possible to find an effect with a single participant and how many trials that requires?

Is the bouba-kiki effect representative?

- The authors' claim refers to sound symbolism in apes as a whole, but they tested a single sound symbolic association (in one ape). There are many non-mutually-exclusive proposed mechanisms for sound symbolism, and it is quite possible that different sound symbolic associations are underpinned by different mechanisms. The authors' proposed mechanism for the bouba-kiki effect (linking it to perception of hand movement while drawing shapes) is not the common explanation for this sound symbolic association. It also doesn't seem to be able to explain the role of vowels in this association. While there is no one-agreed upon explanation for the bouba-kiki effect, it is possible that it is underpinned by associations that apes don't experience (e.g., the round vowels fit the shape of the mouth when producing the vowels but apes don't produce such vowels) whereas other sound symbolic associations (e.g., vowel-size) are underpinned by associations that apes do experience in the world (e.g., larger animals emitting sounds with lower frequencies than small animals). Therefore, at most, this study could suggest that bonobos do not show the bouba-kiki effect rather than that they are not sensitive to sound symbolic associations in general. They might show sensitivity to sound symbolism if tested for sound symbolic associations that are based on co-occurrences that they experience as well.

Appendix B

Associate Editor

Board Member: 1

Comments to Author:

The authors present a study investigating whether a language trained bonobo is sensitive to sound symbolism. The study is likely to generate broad interest and the authors explain its importance to our understanding of the evolution of human language with clarity. Although both reviewers evaluate the study favorably, they raise several elements that will need to be addressed before the manuscript is acceptable for publication. I agree with reviewer 1 that certain aspects of the task may have obscured the intended referential interpretation of the words that were presented. Similarly related to the nature of the task, reviewer 2 raises the concern that the best strategy to solve the test trials may be a 50% decision. Equally important are reviewer 2's comments concerning the absence of priors in the Bayesian estimation (either appropriate priors should be used, or additional justification should be given why default values were used) and the scope of the interpretation of the results.

We thank the editor and the referees for their thoughtful comments and their speedy response. We will answer each of the referee's comments below, including the points raised by the editor. We also added a supplementary file in which we present additional analyses in response to both reviewer's comments. Thank you for helping us improve the quality of the manuscript. Original comments are in black and our responses are in blue and indented.

Referee: 1

The authors' goal is to test whether apes are sensitive to sound symbolism. While they are not the first to test this question, they rightly point out that prior studies might have failed to find such sensitivity because the tested apes did not understand the task, as they were not aware of the potential referential role of sounds. By testing Kanzi, the authors conducted a more informative test of the question. As Kanzi is a unique resource, many would be interested in reading this paper, so it should be published. That said, there are several non-trivial issues with the design which make it difficult to draw clear inferences from the results. The authors should acknowledge and discuss these. There are several additional issues regarding the stimuli, analysis and generalizability of the results to other types of sound symbolism. Some of these can be easily addressed with new analyses. I explain all these points in detail below.

We would like to thank the reviewer for their thoughtful comments. We have carried out a number of additional analyses, as suggested, and have added several paragraphs to explain more thoroughly the experimental rationale, as well as address the valid concerns about generalizability.

Did Kanzi understand and try to engage with the test trials?

- The main contribution of the paper over prior studies that tested sound symbolism in apes is the use of an ape that understands that words are referential. Certain aspects of the task, however, might have obscured this referential interpretation or demotivated Kanzi from engaging with the test trials as he would with regular trials.

o The labels in the test trials were not truly referential, or potentially not referential in a way that Kanzi might understand. That is, whereas in the regular trials each word was associated with a specific object, there was no such relationship between labels and target objects in the test trials. That is, the correct target of a specific word in the regular trials was a specific object, but the correct target of a specific sharp or round word in the test trials was a different shape each time it appeared. Thus, since Kanzi saw the same labels appearing each time with different items, he might have concluded that the sounds are not referential as they don't function the way that other words that he learned function.

o The regular trials used pictures of real world objects. The test trials used abstract drawings. It is unclear whether Kanzi has ever learned to label abstract drawings or understands that words can be referential when there is no real world reference that he can recognize.

We thank the reviewer for pointing this out. Regarding the putatively differing task demands in regular vs. test trials (naming pictures of things that have a name vs. naming abstract drawings for which Kanzi has never learned any names) the goal of embedding a small number of test trials in a large number of regular trials was precisely that the task set "find the best match for the name you just heard" would *carry over* from familiar to unfamiliar stimuli, to approximate as best as possible an "explicit" instruction to match sounds and symbols. Put differently: We know that Kanzi can pair an auditory stimulus (a word) to its correct referent (an object) using a two-alternative forced choice (2AFC) task (see Savage-Rumbaugh et al., 1993; Rabinowitz, 2016). This same paradigm is also one of the most common ways to

explore sound symbolism in humans. Many studies present an auditory stimulus followed by a 2AFC question including a congruent and an incongruent shape (see Maurer et al., 2006; Fort et al., 2015; Margiotoudi et al., 2019; etc.). By placing trials testing for sound symbolism within a match-to-sample task, we hoped to make it clear to Kanzi that his job was to match an auditory stimulus to one of the two available referents: thus, making our task as explicit as possible in a non-linguistic individual. While the reviewer is right in that there are no correct answers in a testing trial compared to a regular trial, this is also true in sound-symbolism tasks in humans. Adults, and even preverbal infants (Asano et al., 2015), form sound-shape associations without being told there is a correct response. If anything, Kanzi was given more direction than human studies, as testing trials were embedded in a match-to-sample task where there was a correct answer.

We apologize that the original manuscript did not make this more clear. We have now added a paragraph at the end of the introduction that explains our experimental rationale more thoroughly. Beyond this clarification, we acknowledge that the reviewer's specific point about Kanzi processing regular vs. test trials differently is valid (also in relation to the differing response time distributions in regular vs. test trials), and will address it in more detail further below.

As for the stimuli, we tried to match the regular trial images to appear similar to the testing trial abstract shapes. All regular trial images were cropped and converted to black and white with a white background. There were also two tokens for each word to prevent a one-to-one matching between word and image. Additionally, Kanzi has very little real-world experience with some of the words (e.g., phone, bubbles, gorilla) and we have no a priori assumption regarding what categories of natural object classes Kanzi thinks should have a name. Much like an abstract shape, a gorilla could be an abstract concept to Kanzi as he has never seen a gorilla in person before. Our study aimed to replicate human studies in a nonhuman animal using stimuli that have been validated and elicit sound-shape associations in humans. Nonetheless, we regard this as a valid point, as outlined below and in one of the newly added paragraphs that addresses this issue in the discussion section.

What we think the experiment succeeds in demonstrating is that even in an ape who successfully matches words to a referent, sound-shape associations are not made when using a task analogous to human tasks where these associations *are* made spontaneously. Put differently, we believe that our sound-symbolism task was as explicit as possible in a non-linguistic individual. That said, we agree with the reviewer to the extent that our task may still not be the best approximation of an explicit test in a nonhuman animal *yet*. For example, a follow up study where

Kanzi is reinforced for a “correct” response, reinforcing the notion that bouba-kiki words *have* referents, could be a next step. Second, testing if Kanzi *learns* congruent word-referent pairings faster than incongruent word-referent pairings would also shed light on the question whether great apes are sensitive to sound-shape associations. For example, a study by one of the present co-authors (Manuel Bohn) found that over time, chimpanzees learned to use information from iconic gestures over arbitrary gestures to correctly locate a reward (Bohn et al., 2016), despite not using this information initially. We also agree with the reviewer that it is possible that over time Kanzi could learn *different* sound-shape associations (other than bouba-kiki-type matchings). We address these different opportunities for future studies in the two new paragraphs of the discussion section.

o Kanzi did not receive any feedback, including not receiving a reward. The authors explain that this was intended to prevent learning. This is understandable (though sound symbolism is often tested with learning paradigms). Nevertheless, as the trials were clearly distinguishable from the others (unfamiliar words, abstract drawings etc.), Kanzi could have quickly realize that there is no point in putting in the effort as he would not be rewarded and would not even know if he were correct.

o Indeed, the completely different RT distribution for test and regular trials suggests that Kanzi did not attempt to solve the test trials but just respond quickly and move on. The authors state that shorter RTs do not indicate less effort because, in the regular trials, shorter RTs were associated with more correct responses. The differences are not the same though. The shape of the RT distributions of the correct and incorrect regular trials are identical, only slightly shifted. The RT distribution of the test trials is different, extremely narrow, and shifted to a large degree. It very much looks like a floor effect of responding as quickly as possible.

We thank the reviewer for pointing this out. We think this interpretation is important to consider and have adjusted the text in the result section accordingly, and have added an in-depth discussion to the Discussion section. Regarding the reviewer’s first point, that Kanzi may have “quickly realized” a qualitative difference between test and regular trials (unfamiliar words, abstract drawings) and subsequently shifted back and forth between “effortful responding” and “random responding”, depending on trial type: we have now formally tested for this possibility by looking at how RTs for each trial type developed over time, that is across trials. As readers of the manuscript will now see in Figure 3 of the Supplementary Material and the

accompanying text, there was no effect of Kanzi becoming faster / less effortful in test trials *to a greater extent* than he did in regular trials. That is, if Kanzi did “realize” the difference between the two trial types, this must have happened practically instantaneously. Additionally, the lack of reward alone may also be insufficient in explaining why Kanzi would see “no point in putting in the effort” as even correct performance in regular trials also often remained unrewarded (and in sum, more often even than the number of all test trials).

That said, we concur with the reviewer that the difference in RT distributions demands a qualification in the conclusions that may be drawn from Kanzi failing to show a bouba-kiki effect in our paradigm. As we have written above, our goal here was to present Kanzi with stimuli that have been validated, and shown to elicit sound symbolic effects, in humans, all embedded in an MTS task that Kanzi already knows. The tradeoff is that, indeed, there are the qualitative differences between regular and test trials that the reviewer points out. To address this, we discuss the issue in an additional paragraph in the discussion section and include a suggestion how the addition of certain trials in future studies (e.g. novel English words with novel pictures that have a clear, rewarded solution) could further improve the likelihood that the implied task demands - find the picture that matches the word - *carry over* from regular MTS trials to bouba-kiki trials.

Stimuli

- The authors used the stimuli from Margiotoudi et al. (2019). While the prior study found an effect with humans with these stimuli, it is not fully clear how the supposedly sound symbolic words were generated. In Margiotoudi et al. (2019) the authors mentioned relying on prior studies, but examination of those studies does not align well with the generated words. For example, Margiotoudi et al. (2019) cite McCormick et al. (2015) as the basis for their phoneme selection, but three of the five phonemes that were used to create sharp words (/s/, /f/, and /z/) were rated as in-between sharp and round in McCormick et al. (2015). In fact, their ratings were very similar to those of /d/, which was used as a round phoneme. It might be better to run an additional analysis of the data, including only words constructed with phonemes that prior studies suggest are sound symbolic, namely, the subset of sharp words with /k/ and /p/ and the subset of round words with /l/, /m/, and /n/.

Thank you very much for this comment. We have now included the analysis suggested by the reviewer in the supplementary material, and refer to it in the

manuscript. When we fit our model to a subset of the data only including test trials with words that received high ratings in McCormick et al. (2015), we get a nearly identical result compared to the analysis including all trials (see Supplementary Figure 7).

Analysis

- The authors stated that they used the default values in their Bayesian analysis. This is not recommended. It matters greatly which priors are chosen and they should be theoretically and/or empirically motivated (e.g., consider the effect size in humans and the difference between effects sizes in humans vs apes). For example, selecting too large of an expected effect size will show a null effect even when there is a difference, just not as large as the default setting prescribes. While the plot suggests there really is no effect, the test should be done with appropriate parameters. This will become important if additional analyses are added (e.g., on a subset of the words – see comment above).

We completely agree that the choice of priors is an important issue in Bayesian analysis. Indeed, the choice of priors would be very important if our inference would have been based on a model comparison via Bayes Factors because Bayes Factors are computed based on the likelihood of the data under each model across the prior. However, in our case, the choice of priors is less likely to matter, because our inference is based on the posterior estimate (with 95%CrI) of the intercept term. Given the large amount of data that we have to estimate this parameter, any prior is likely to be “washed” out, that is, unless the prior prohibits sampling in a certain range, the posterior distribution will always end up in the same region. We confirm this is in a prior sensitivity analysis now included in the Supplementary material: No matter how strong a prior we choose for the intercept term, the posterior estimates are always identical. This is even the case when we choose a prior that assumes a very strong effect (~73% correct responses). Figure 6 in the Supplementary Material visualizes these results and is now referenced in the manuscript.

- - Have the authors conducted any power analysis to determine whether it is possible to find an effect with a single participant and how many trials that requires?

Thank you very much for bringing this up. We did indeed conduct a power analysis for this purpose ahead of time, and referred to it in the pre-registration of the study. The corresponding analysis code can be found in the online repository associated with the study and we now include a brief description of it in the Supplementary Material (Figure S1, which is now also referenced in the manuscript). In short, when assuming a true effect of 65% correct responses with 300 test trials our model would support the conclusion that performance was above chance in 88/100 cases (roughly power of 0.88). (The task was designed as an “explicit” task in which humans responded correctly in ~ 70% of cases. To be on the safe side we did our power analysis assuming 65% correct responses.)

Is the bouba-kiki effect representative?

- The authors' claim refers to sound symbolism in apes as a whole, but they tested a single sound symbolic association (in one ape). There are many non-mutually-exclusive proposed mechanisms for sound symbolism, and it is quite possible that different sound symbolic associations are underpinned by different mechanisms. The authors' proposed mechanism for the bouba-kiki effect (linking it to perception of hand movement while drawing shapes) is not the common explanation for this sound symbolic association. It also doesn't seem to be able to explain the role of vowels in this association.

While there is no one-agreed upon explanation for the bouba- kiki effect, it is possible that it is underpinned by associations that apes don't experience (e.g., the round vowels fit the shape of the mouth when producing the vowels but apes don't produce such vowels) whereas other sound symbolic associations (e.g., vowel-size) are underpinned by associations that apes do experience in the world (e.g., larger animals emitting sounds with lower frequencies than small animals). Therefore, at most, this study could suggest that bonobos do not show the bouba-kiki effect rather than that they are not sensitive to sound symbolic associations in general. They might show sensitivity to sound symbolism if tested for sound symbolic associations that are based on co-occurrences that they experience as well.

We thank the reviewer for this comment. We clarified throughout the manuscript that we refer to this specific type of sound symbolic association (i.e., pseudoword-shape) and not to all the other cases covered by the term of sound symbolism. The new title for the manuscript now also acknowledges this distinction. We acknowledge also in the discussion that the proposed mechanism of hand action knowledge can provide a plausible explanation only for this specific type of sound

symbolism (pseudoword-shape associations) and not other types of sound symbolism (e.g., pitch-size).

At this point, we would like to clarify that in the hand action knowledge account for sound symbolic associations of pseudoword-shape types, the role of vowels and consonants is equally important. In this account, the 'bouba' and 'kiki' type of pseudowords resemble the sounds of our 'round' or 'sharp' hand actions respectively. The human speech sounds we use to imitate the acoustic products of these 'round' and 'sharp' hand actions can consist of both vowels and consonants. Hence, the hand action knowledge does not aim to provide an explanation on the importance of either vowels or consonants in the bouba-kiki effect, but explain why we associate 'round' and 'sharp' sounding pseudowords with curved and edgy abstract shapes.

We agree with the reviewer that other accounts have been proposed to explain the bouba-kiki effect, such as the well-known synesthetic-articulatory account of sound-shape association (Ramachandran & Hubbard, 2001). This latter account could partially explain the absence of sound-shape associations in great apes. As mentioned by the reviewer the apes cannot produce rounded vowels. The articulatory account could provide an explanation for the bouba-kiki effect only for the rounded vowels that are often observed in 'round' sounding pseudowords (e.g., Maurer et al., 2006) but not for other types of vowels or consonants found in pseudoword-shape sound symbolic mappings (e.g., Fort et al., 2015, for a discussion, see Margiotoudi and Pulvermüller, 2020). On the other hand, the hand action knowledge account identifies the mechanism of the bouba-kiki effect to the knowledge of the visual, acoustic and motor products of our actions, without aiming to answer the question on whether vowels or consonants make a pseudoword more 'round' or 'sharp' sounding. The latter account proposes that the knowledge of the multimodal products of our actions may set the stage for the emergence of bouba-kiki associations. Based on studies on comparative neuroanatomy between human and non-primates, there is evidence for a lesser white matter connectivity in the brains of great apes on neuroanatomical tracts (e.g., the arcuate fascicle) that connect modality specific and multimodal areas and are relevant in sensorimotor binding (De Schotten et al., 2012; Rilling et al., 2008). A weaker connectivity across sensory and motor brain regions could not facilitate the storage of a hand action sound-shape knowledge learned by statistical co-occurrences (i.e., co-occurrence of visual, acoustic and motor products). Consequently, according to the hand action account, great apes cannot be sensitive in detecting bouba-kiki type associations. Of course, even if a previous study has provided the first experimental evidence on the relationship between action knowledge and bouba-kiki associations (Margiotoudi and Pulvermüller, 2020), the mechanism

behind bouba-kiki type associations in humans - and why other apes may lack it - requires further investigation. Please let us know if you believe this section needs further qualifications or additions.

Finally, we agree with the reviewer that great apes and other non-human primates might be sensitive to other sound symbolic associations (e.g., pitch/vowel-size) because they are exposed to these statistical co-occurrences in their natural environment, but also have the necessary neuroanatomical infrastructure that allows the statistical learning between certain modality-specific features. For example, great apes could be sensitive to pitch size associations a) because they encounter these type of associations as described by the reviewer (frequency code: Ohala, 1994), 2) because they show an expanded occipito-temporal white matter tract connectivity similar to humans that would permit the associative learning between 'low-level' perceptual mappings (e.g., pitch & size) (Roumazeilles et al., 2020). We acknowledge the need for additional investigations into other sound-symbolic associations in great apes in the discussion.

Referee: 2

Comments to the Author(s)

The study „Bo-NO-bouba-kiki: Picture-word mapping but not sound symbolism in a language trained bonobo“ is in general a well-designed study asking whether the language trained bonobo Kanzi is able to use sound symbolism in a matching-to-sample task. The methodological design and the results are well presented. I have only one major comment.

In total Kanzi completed 3000 trials, 2700 regular trials and 300 test trials. In the 2700 regular trials the task was to match English sounds like apple or knife to the correct picture, independent of any sound symbolism. That means that Kanzi must remember the correct picture of certain sounds. It could be that this sound/ picture pair followed sound symbolism rules or not. But important to solve the task was that Kanzi had this sound/ picture pairs in his memory. In one of ten cases Kanzi get a sound with an unknown picture. My question is: Why should Kanzi come up with the idea that using sound symbolism is a good strategy to solve this task. In case that all known pairs followed a sound symbolism rule this could be a good strategy, but the authors used although a lot of pairs which did not followed such a rule. Therefore, a 50% decision in case Kanzi did not know the sound/ picture pair seems to me the best possible strategy.

We thank the reviewer for this comment. In part it overlaps with the point by reviewer 1 about the difference in task demands in regular vs. test trials. As we have stated above, we have now addressed the issue by explaining our rationale in more detail in the introduction, and by addressing these differences in the limitations section, adding a suggestion for how future studies may still be able to find a bouba-kiki effect with adjusted paradigms. We have also added a number of additional analyses (see above) to investigate whether Kanzi approached these trials with different strategies.

In addition, we would like to clarify in case there is a misunderstanding here: the goal of our study was not to *prime* sound-symbolic responding, as it were, by rewarding sound symbolic matching in regular trials, and to see if an already reinforced bouba-kiki-type-matching-strategy would carry over to abstract test trials. Rather, the testing rationale was to use the matching-to-sample task with English words that Kanzi was already familiar with, to prime an expectation in regular trials that “the best answer is to pick the picture that matches the word” and see, if forced to apply this strategy to new words and pictures (= the test stimuli), if Kanzi would offer spontaneously sound-symbolic matchings as his “best guess”, akin to humans showing sound-symbolic matching intuitively when given the same unfamiliar stimuli. The aforementioned paragraph in the introduction makes this more explicit now. We hope that we were able to strike the right balance.

Minor comment:

I think it good be helpful for some readers if the authors give a short explanation, why the used Bayesian Generalized Linear Mixed Model.

Thank you for pointing this out. The main reason for choosing Bayesian instead of frequentist models was that frequentist models often run into convergence issues. In a Bayesian framework this is not an issue, which guaranteed that we could run the models we pre-registered and on which our power analysis was based. The general model architecture would have been the same in a frequentist framework.

References

Asano, M., Imai, M., Kita, S., Kitajo, K., Okada, H., & Thierry, G. (2015). Sound symbolism scaffolds language development in preverbal infants. *cortex*, 63, 196-205.

Bohn, M., Call, J., & Tomasello, M. (2016). Comprehension of iconic gestures by chimpanzees and human children. *Journal of experimental child psychology*, 142, 1-17.

de Schotten, M. T., Dell'Acqua, F., Valabregue, R., & Catani, M. (2012). Monkey to human comparative anatomy of the frontal lobe association tracts. *Cortex*, 48(1), 82-96.

Fort, M., Martin, A., and Peperkamp, S. (2015). Consonants are more important than vowels in the bouba-kiki effect. *Language and Speech*, 58(2):247–266.

Margiotoudi, K., Allritz, M., Bohn, M., & Pulvermüller, F. (2019). Sound symbolic congruency detection in humans but not in great apes. *Scientific reports*, 9(1), 1-12.

Margiotoudi, K., & Pulvermüller, F. (2020). Action sound–shape congruencies explain sound symbolism. *Scientific reports*, 10(1), 1-13.

Maurer, D., Pathman, T., and Mondloch, C. J. (2006). The shape of boubas: Sound–shape correspondences in toddlers and adults. *Developmental science*, 9(3):316–322.

Moseley, R. L., & Pulvermüller, F. (2018). What can autism teach us about the role of sensorimotor systems in higher cognition? New clues from studies on language, action semantics, and abstract emotional concept processing. *Cortex*, 100, 149-190.

Occelli, V., Esposito, G., Venuti, P., Arduino, G. M., & Zampini, M. (2013). The Takete—Maluma phenomenon in autism spectrum disorders. *Perception*, 42(2), 233-241.

Ohala, J. J. (1994). The frequency code underlies the sound-symbolic use of voice pitch. *Sound symbolism*.

Rabinowitz A. (2016). Linguistic competency of bonobos (*Pan paniscus*) raised in a language-enriched environment. Master's thesis, Iowa State University, Ames, IA. See <https://lib.dr.iastate.edu/etd/15794>

Ramachandran, V. S., & Hubbard, E. M. (2001). Synaesthesia--a window into perception, thought and language. *Journal of consciousness studies*, 8(12), 3-34.

Rilling, J. K., Glasser, M. F., Preuss, T. M., Ma, X., Zhao, T., Hu, X., & Behrens, T. E. (2008). The evolution of the arcuate fasciculus revealed with comparative DTI. *Nature neuroscience*, 11(4), 426-428.

Roumazeilles, L., Eichert, N., Bryant, K. L., Folloni, D., Sallet, J., Vijayakumar, S., ... & Mars, R. B. (2020). Longitudinal connections and the organization of the temporal cortex in macaques, great apes, and humans. *PLoS biology*, 18(7), e3000810.

Savage-Rumbaugh, E. S., Murphy, J., Sevcik, R. A., Brakke, K. E., Williams, S. L., Rumbaugh, D. M., and Bates, E. (1993). Language comprehension in ape and child. *Monographs of the society for research in child development*, i-252.

Appendix C

Associate Editor

Board Member: 1

Comments to Author:

The authors completed an in-depth revision in response to the reviewer's comments and I believe this has further improved the quality of the paper. I do agree with both reviewers, however, that some minor, yet crucial nuances should still be clarified before moving towards publication. Reviewer 1 notes the absence of an explicit acknowledgment that sound symbolism could occur if Kanzi were confronted with a sound symbolic association that would fit with his experience in the world. Reviewer 2 notes that the clarification that Kanzi showed no spontaneous sound symbolic matching should be noted earlier in the manuscript.

We thank the editor and the referees for their time in reviewing our manuscript and for their thoughtful comments towards improving the quality of our manuscript. We have made the suggested changes in the title, abstract, and in the discussion section. We answer below both points highlighted by the editor and the reviewers. Original text is in black and our responses are in blue and indented.

Reviewer(s)' Comments to Author:

Referee: 1

Comments to the Author(s)

The authors addressed many of my prior comments in detail. I appreciate the new analyses in the appendix. At the same time, there is still one issue that I think is not addressed sufficiently. In particular, the sound-shape association is one type of sound symbolic association, and not necessarily a representative one in terms of mechanism. Therefore, it is quite possible that Kanzi would not exhibit sound-shape sound symbolism but would exhibit other types of sound symbolism (such as pitch-size). The authors addressed my prior comment by changing the title and in their phrasing in some locations. At the same time, they never explicitly acknowledge the fact that it is possible that had they tested Kanzi with a different sound symbolic association, he would have exhibited sound symbolism. That is, the problem might be due to the fact that this specific association is not one that fits with his experience or that he can represent, but that he would be sensitive to sound symbolism that fits with his experience in the world.

Thank you very much for this comment. As we mentioned in the first revision, we agree completely that Kanzi might be sensitive to other types of sound symbolic mappings. We added a paragraph in the discussion, where we acknowledge the possibility that Kanzi could be sensitive to other types of mappings (e.g, pitch-size), which match better his individual experience.

Another more minor issue is that when discussing differences between the two tasks, it is worth mentioning that Kanzi was rewarded for his responses on the regular trials, but not on the test

trials. This could have influenced his motivation to respond in the test trial and explain the fast and chance responses. The authors responded that he wasn't rewarded 100% of the time in the regular trials. True, but he was rewarded on the majority of them (67%). Had they thought that rewards would not motivate behavior, reward probably wouldn't have been included at all.

We have added an explicit acknowledgement of the difference in reinforcement schedule to the part of the discussion section where we discuss differences between stimuli in regular and test trials.

To be clear, we acknowledge, both in this response, as well as the manuscript, the possibility that Kanzi could have used different strategies between regular and test trials, as potentially suggested by the difference in RT distributions between the two conditions. As we also mention in the discussion section, however, we don't believe it to be particularly plausible that a difference in strategy (as it may be suggested by differing response time distributions) would be caused by the differing reinforcement schedules. We don't believe this to be particularly plausible because the fact that test trials were not reinforced do not make them "stand out" (in comparison to regular trials which outnumbered test trials 9:1 and were also often not reinforced), or at least it does not make them stand out nearly as much as the difference in content. We understand that the concern here may be that Kanzi would learn over time, from differing reward schedules alone, that there is "no point" in looking for a correct choice in test trials and default to a random choice strategy whenever he encounters one of these special trials. However, we believe that the lack of difference in RT decline between regular and test trials suggests that the reinforcement schedule does not account for the potential difference in strategy. That said, we agree, of course, that there was a reason for providing reinforcement on regular trials, both to keep Kanzi motivated to participate as well as to create expectations that there always is a "correct choice" to be made. We hope that with the explicit acknowledgement of these differences in the manuscript and the discussion of response times we were able to strike the right balance for readers.

Referee: 2

Comments to the Author(s)

The authors made an extensive revision and discuss carefully my concerns. The authors added several paragraphs in the introduction and discussion to discuss the problematic with differences in task demands (regular vs. test trials), and added a number of additional analyses to investigate whether Kanzi approached these trials with different strategies. They specified that their research question, whether Kanzi would offer spontaneously sound-symbolic matchings as "best guess", akin to humans showing sound-symbolic matching intuitively when given the same unfamiliar stimuli.

I think it would nice to find this clarification that Kanzi showed no "spontaneous" sound symbolic matching either in the abstract or title.

Thank you very much for this comment. We added the word "spontaneous" both in our title and abstract.